# A Method to Estimate Horse Speed per Stride from One IMU with a Machine Learning Method

**DOI:** 10.3390/s20020518

**Published:** 2020-01-17

**Authors:** Amandine Schmutz, Laurence Chèze, Julien Jacques, Pauline Martin

**Affiliations:** 1Lim France, Chemin Fontaine de Fanny, 24300 Nontron, France; pmartin@lim-group.com; 2CWD-Vetlab, Ecole Nationale Vétérinaire d’Alfort, F-94700 Maisons-Alfort, France; 3LBMC UMR T9406, Université de Lyon, Lyon 1, 69364 Lyon, France; laurence.cheze@univ-lyon1.fr; 4ERIC EA3083, Université de Lyon, Lyon 2, 69007 Lyon, France; julien.jacques@univ-lyon2.fr

**Keywords:** speed estimation, support vector machine, overall dynamic body acceleration, sensors, horse

## Abstract

With the emergence of numerical sensors in sports, there is an increasing need for tools and methods to compute objective motion parameters with great accuracy. In particular, inertial measurement units are increasingly used in the clinical domain or the sports one to estimate spatiotemporal parameters. The purpose of the present study was to develop a model that can be included in a smart device in order to estimate the horse speed per stride from accelerometric and gyroscopic data without the use of a global positioning system, enabling the use of such a tool in both indoor and outdoor conditions. The accuracy of two speed calculation methods was compared: one signal based and one machine learning model. Those two methods allowed the calculation of speed from accelerometric and gyroscopic data without any other external input. For this purpose, data were collected under various speeds on straight lines and curved paths. Two reference systems were used to measure the speed in order to have a reference speed value to compare each tested model and estimate their accuracy. Those models were compared according to three different criteria: the percentage of error above 0.6 m/s, the RMSE, and the Bland and Altman limit of agreement. The machine learning method outperformed its competitor by giving the lowest value for all three criteria. The main contribution of this work is that it is the first method that gives an accurate speed per stride for horses without being coupled with a global positioning system or a magnetometer. No similar study performed on horses exists to compare our work with, so the presented model is compared to existing models for human walking. Moreover, this tool can be extended to other equestrian sports, as well as bipedal locomotion as long as consistent data are provided to train the machine learning model. The machine learning model’s accurate results can be explained by the large database built to train the model and the innovative way of slicing stride data before using them as an input for the model.

## 1. Introduction

According to Article 234 of the International Equestrian Federation (FEI) Jumping Rules, horses speed for international competitions has to be 350 m per minute at a minimum and 400 m per minute at a maximum, with exceptions for different kinds of show conditions (FEI, FEI Jumping Rules, 26th edition, 2019). Speed is therefore a key parameter for success in show jumping competitions and an important training input. 3D optical motion capture is currently the gold standard for horse gait analysis and can be therefore used for measuring stride parameters such as speed [1]. However, the setting up of its measurement field is time consuming, as well as the data processing when your subject differs from the plug-in gait reference provided by the software [2]. This leads to an impossible use on a daily basis or during championships for a rider who wants descriptive results of his/her horse’s performance and locomotion parameters within a minute or even potentially in real time without preliminary preparation.

New gait analysis techniques emerged and enabled the development of tools to provide objective parameters of horses’ motion [3] or to detect lameness [4], using low cost inertial measurement units (IMU), composed of two sensors: tri-axial accelerometer and tri-axial gyroscope. Those sensors can be coupled with a tri-axial magnetometer and are therefore called mIMUs. Thanks to data fusion techniques, the use of a magnetometer helps reduce the IMU bias and leads to better estimation of distance [5]. IMUs can also be paired with a global positioning system (GPS) unit, to improve the estimation of locomotion parameters such as speed [6,7]. However, GPS can be badly influenced by the presence of obstacles [8], and it cannot be used indoors due to signal loss under roofing.

There exist three main families of methods developed to calculate motion characteristics from IMU signals. Firstly are the model based methods, like inverted pendulum models for speed estimation in human gait, which simplify complex biomechanical behaviors with a simple mechanical model and incorporate subject specific information like the limb length [9,10]. Secondly are the signal based methods, which mainly rely on signal integration [11] and use signal processing methods like the Butterworth filter to prevent drifting [12]. Those methods need to formulate some realistic assumptions to correct sensors’ drift and need a zero velocity phase within each stride to be able to apply the integration process. For example, the method proposed by [1] estimated horse displacement from one IMU placed on the trunk, assuming that the horse was in a steady state because of a treadmill that constrained the horse’s motion. In this case, the IMU sensor displacement should follow a closed loop, and then, the average velocity over a stride should be zero, as well as the average forward-backward and side-to-side acceleration. Thus, in this context, stride-by-stride mean subtraction of acceleration and of the calculated velocity before integration enables determination of the integration constants. This assumption is often invalid in numerous experimental conditions, leading to the non-applicability of the direct signal integration method.

Thirdly, new methods based on statistical approaches are developed to estimate human speed [13,14] from IMU data. Those approaches provide accurate estimation of walking speed, but the regression models’ accuracy seems to be dependent on the range of motion. To prevent the drift, model extension is proposed to divide data according to the speed regime prior to the speed estimation [15]. Then, a regression model is fit independently to each range of speed. A support vector machine (SVM) [16] is used for the classification part. This method refines the regression model’s accuracy for the slow speed regime. SVM is a machine learning approach that can be used for both classification and regression. The concept is simple: one has to provide a dataset, called the training dataset, with a known variable of interest value (for example, IMU signals matched to their associated speed) that will be used to build a model. The model will then be able to predict the value of the variable of interest for new data. The model has to be trained with cases that can be encountered in its future application, without which it will perform poorly.

The objective of this work is to develop a model that can be included in a smart device that can provide the rider with the movement parameters of his/her horse, in daily routines such as during a training session, as well as during competition events, using only one IMU fixed in the pommel of the saddle. This user case differs from existing published work for sports [17] by not using sensor fusion, not being in a steady state that allows an easy use of direct integration of acceleration signals, nor using a sensor on the limb, which allows resetting errors at each cycle over short time periods. A new model, based on the SVM regression method, to predict horse speed at canter from one IMU’s data only, will be presented. The idea is also to propose a tool that overcomes the limits imposed by the use of GPS or 3D optical motion capture systems. The accuracy aim for the developed model is 0.6 m/s (36 m/min) in order to meet the expectations of the show jumping professionals. As far as the authors know, this accuracy was not reached for horses by the previous mentioned [1,12] methods using data from one IMU only. The results obtained with this new model will be compared to those of one signal based method, already used for speed estimation in animal locomotion, the overall dynamic body acceleration.

## 2. Materials and Methods

### 2.1. Data

The database used for model development was made of 3221 canter strides from 58 ridden jumping horses of different breeds, size (129–176 cm), age (5–18 years old), and different levels of competition (amateur or professional).

One IMU (LSM6DSL, STMicroelectronics, Geneva, Switzerland) placed in the saddle pommel close to the horse’s withers was used to measure tri-axial acceleration (range ±8 g) and tri-axial rotation rate (range ±2000 dps) at a sampling frequency of 100 Hz. It was fixed directly to the saddle tree, before its assembly. Data collected by the IMU were sent via a Bluetooth® antenna to a smartphone (iPhone X, Apple Inc., Cupertino, CA, USA) and then stored on an online server.

Two different protocols were used to collect data: the first one was speed measurement for a straight path, and the second one was speed measurement for a curved path. For both protocols, reference speed was measured by video cameras or a chronometer and matched to each stride signal to build the machine learning model. In practice, a stride is defined as the period between two successive hoof falls of the same leg. For the present work and each protocol, “strides” are defined from the maximum peak on the *Z*-axis, which corresponds to the dorso-ventral axis, (cf. Figure 1) of the raw acceleration data to the next 100 samples, in order to have the same number of points for each individual regardless of the speed, a necessary condition to use machine learning methods. Therefore, depending on the horse’s speed, this data segmentation may include in one “stride” more than one real stride. The authors chose not to re-sample a cycle in order to keep the information on the duration of the stride to estimate speed. Values from the three axes of the gyroscope and the accelerometer were extracted according to this cutting with an automated detection algorithm written with MATLAB (The MathWorks, Natick, MA, USA). Therefore, one “stride” data were 6 × 101 samples (101 values for 1 s of measurement and 1 column per axis).

#### 2.1.1. Straight Path

To get reference stride speed, IMU data were synchronized to a 4 camera 2D tracking system (Imaging Development System, GmbH, Obersulm, Germany), which had a measuring field of 26 m. Horses were equipped with 10 2D reflecting markers on anatomical landmarks (Figure 2), and their speed in the camera’s field was derived from the markers’ 2D trajectories using a custom software written with MATLAB® (R2014b). The accuracy of this system was 1.4% of the measured distance [18], which would correspond to ±2.8 cm for a 2 m measured distance, for example.

Data were gathered for different speeds chosen by the rider (normal, slow, and fast), with and without jumps, before and after the camera’s field, and with or without ground bars, spaced from 2.5 to 4.5 m, in the field of measurement. Those various conditions were chosen in order to expand the range of canter within the training set, in order to get closer to the daily training conditions of jumping horses.

#### 2.1.2. Curved Path

Because the 2D tracking system had great accuracy only when the horse displacement was perpendicular to the cameras field, another measurement protocol was designed for curve displacement.

A curved path of a known perimeter was defined with cones and with a width small enough to limit the horse’s possible pathway (Figure 3). The traveled distance was calculated as distance = 2πr, with *r* the radius of the circle.

Time spent in the curve by the horse was measured with an automatic chronometer (CP 520, Tag Heuer) triggered at the entrance and at the exit of the curved path (Figure 3). The average speed of the horse was then derived as speed = distance/time. Each stride of the horse within the curve was then matched with the average speed. For example, if the average speed in the curve was 6 m/s and the horse did 5 strides, then a speed of 6 m/s was assumed for all these strides.

In order to mimic real-life conditions of a jumping course, the whole database was composed of 2906 strides collected in a straight path and 315 strides in a curved path. An example of signals for one stride is available in Figure 4. The red curve corresponds to a running speed of 8.6 m/s, and the black one corresponds to a running speed of 5.0 m/s. The signal length for one stride was correlated to the speed.

### 2.2. Speed Measurement Methods

#### 2.2.1. Overall Dynamic Body Acceleration Method

The overall dynamic body acceleration (ODBA) method is a signal based method proposed in [19] that does not rely on signal integration. The authors developed a parameter named ODBA, calculated from acceleration in the 3 space directions, which was closely linked to the speed of a walking animal.

In this case, acceleration signals were low pass filtered using a fourth-order Butterworth filter with a cut-off frequency of 10 Hz. After that, an angle correction was applied to align the *Z*-axis with the gravity vector. Then, for each axis, as specified by [19], the signal mean value was subtracted to smooth the data. Those values were then converted to absolute positive units. Finally, the resulting signals were summed up, and a mean ODBA value was calculated for each stride. A linear regression was then used to link the mean ODBA for one stride to the speed of the stride.

The linear regression was performed with R software (v.3.4.0, Vienna, Austria) [20] and the lm function of the stats package [20].

#### 2.2.2. Machine Learning Method

Several methods can be tested in order to predict the speed from the IMU signals, such as Ridge, Lasso, Partial Least Squares (PLS), Principal Component Regression (PCR) and elastic net regressions [21], random forests, neural networks, SVM [16], and non-parametric [22] and parametric functional regression [23]. All those methods have been tested on the current database, so we will only present here the one that gave the best results.

In the present work, the SVM method was the one that gave the best results. SVM is a supervised learning algorithm that can be used for both classification and regression [16]. Its goal is to find a function f(x) that has at most ϵ deviation from the actually obtained target $y_{i}$ for the training data (here, the speed per “stride”) and at the same time is as flat as possible (in the present case, the smoothness parameter was fixed to 4). Thus, the model did not consider errors as long as they were less than ϵ, but would not accept a deviation larger than this. The proportion of the training set used to create support vectors was also a tuning parameter of the algorithm, and it was limited to 75% in the present work, then ϵ was automatically calculated by the algorithm [24].

To sum up, the six signals collected with the accelerometer and the gyroscope for each “stride” were matched with the measured reference speed for the stride. This was used as input data to train the SVM model in order to obtain the best speed estimation for new data in the future. All this process is illustrated in Figure 5. The model was developed with R software and the svm function of e1071 package [25].

#### 2.2.3. Methods Comparison

To compare the accuracy of both the ODBA and SVM models, the database was cut into 2 parts: a training dataset, which was composed of a random sampling of 80% of the database, and the remaining 20% forms the test data set. This sampling prevented over-fitting because the same stride could not be in both sets. The 2 models were built on the training dataset, and their accuracy was evaluated on the test dataset.

Comparison between models was done with the calculation of the percentage error in the estimated speed above 0.6 m/s. This threshold was the minimum satisfactory for this parameter to make sense for the professionals. Percentage error above the target value of 0.6 m/s was computed as: $$\begin{array}{c} \%error\;above\;target\;value=100\times \sum_{i}\frac{|Measured\;speed\;at\;stride\;i-Predicted\;speed\;at\;stride\;i|>0.6}{Total\;number\;of\;strides}, \end{array}$$
with *i* corresponding to each stride of the test dataset.

Models accuracy was also compared with:$$\begin{array}{c} RMSE=\sqrt{\sum_{i}\frac{{\left(Measured\;speed\;at\;stride\;i-Predicted\;speed\;at\;stride\;i\right)}^{2}}{Total\;number\;of\;strides}} \end{array}$$
and Bland and Altman plots and its 95% limits of agreement [26], which allowed the evaluation of differences between 2 methods used on the same individuals (here, strides). In this work, the average differences between each method and reference values obtained with 2D tracking system for straight path and chronometer for curve path were examined. Bland and Altman analysis and graphs were built with the bland.altman.plot function from the BlandAltmanLeh R package [27].

## 3. Results

To avoid results’ fluctuation due to random sampling of the test set, the random sampling process of the database was repeated 50 times, and the average, minimum, and maximum of percentage error were estimated for each repetition, as well as the width of the Bland and Altman limit of agreement.

Table 1 shows the mean results for each model. With an average percentage of error of 10.9% against 51.4% for the ODBA method, the SVM model clearly outperformed its competitor.

The Bland and Altman plot of one SVM repetition is shown in Figure 6 (top), with one circle corresponding to one stride. The speed predicted by the model and the measured speed of the stride were compared. The mean bias was zero, which meant that on average, the SVM model output was close to the measured speed. If the model predictions were perfect, all the points would be aligned on the zero line. The points that were the farthest from the zero line were the worst predictions. We can see that for some strides of low speed (below 5 m/s), the SVM model had a tendency to overestimate their speed; whereas for some strides of high speed (above 5 m/s), the SVM model had a tendency to underestimate them. Nevertheless, 95% of strides had a bias lower than 1 m/s, which was satisfying according to this work’s objective to reach an accuracy of 0.6 m/s; whereas ODBA estimations (Figure 6, bottom) were more variable than SVM ones. The mean bias was also zero, but the 95% confidence interval was twice the size of the SVM one (cf. Table 1), that is to say high above our objective value. The ODBA method was more variable than the SVM one, with 95% of strides’ bias lower than 2.5 m/s and a clear tendency to underestimate strides of high speed.

## 4. Discussion and Conclusions

The objective of our study was to develop a model that could be included in a smart device in order to provide horse speed per stride from only one IMU situated on the horse wither. The number of sensors was reduced to a minimum in order to facilitate the daily use of this tool, and the non-use of GPS was due to our willingness to make this tool work both inside and outside.

Usually, models for speed estimation first detect a stride, then cut the collected signal according to this stride and apply the calculation model [28,29,30]. These methods standardized the stride duration for all individuals, and few of them took into account the stride duration to calculate the speed parameter leading to a loss of information. In the present work, the authors chose a different way of pre-processing the collected signals. Precisely, the stride was detected on the *Z*-axis (dorso-ventral axis), but 101 points were kept from the maximal peak on the *Z*-axis. This change allowed keeping the stride duration information: for high speed, the “stride” would have a signal that contained more canter cycles than low speed “strides”. This cutting helped increase the accuracy of the machine learning model. Indeed, the presented model applied to data sampled on real strides showed the average percentage of errors above 0.6 m/s increase by 2%.

To evaluate the presented SVM method, its results were compared with those of another method that did not need external inputs to estimate speed per stride from new IMU data: the ODBA method. In fact, direct signal integration methods, which are commonly used in the human case or in the horses running on a treadmill case, need strong assumptions to calculate an integration constant for speed estimation, which cannot be made when the horse moves in real conditions and when the IMU is not located on the limb [5,17]. Moreover, biomechanical models have not been developed for an asymmetrical gait such as horse canter.

The novelty of this paper was to propose a model for speed estimation that relied on one IMU only. The integration of the machine learning model in a device for equestrian sports was innovative in comparison with other existing systems for equestrian sports based on GPS or in comparison with human tracking motion systems that were mainly based on the use of a magnetometer or several IMUs [5]. The machine learning approach allowed the development of a smart device that did not rely on a GPS for the estimation of a physical phenomenon, here the horse speed at each stride, with an average accuracy of 0.6 m/s. In fact, for one jumping course of 250 strides, 223 strides would be estimated with an error lower than 0.6 m/s. This accuracy met the expectations of professionals in the show jumping discipline, which was their main concern about using or not connected devices. As expected, as show jumping can be practiced both indoors and outdoors, our tool overcame the GPS system’s limitations.

The presented model cannot be benchmarked to other works on horses because no one else has provided a speed per stride estimation. As a matter of fact, the work in [1,12] calculated traveled distance with preciseness, but the work in [12] aimed to provide a speed estimation in future work. In human research, a wide literature exists on computing human walking speed from data collected by one IMU placed on the foot, as for example [29], who compared two methods of walking speed estimation, whose ARMSE range was 0.2–0.3 km/h (0.06–0.08 m/s) depending on the walking speed and the method used. The work in [13] estimated an instantaneous velocity decomposed in the three space directions from two IMUs’ data placed on the pelvis and on the shank of the subject, whose accuracy was in the same range as the previous study. The work in [31] developed a model for walking speed estimation based on a regression model, which used data from one wrist worn inertial sensor. In their paper, the Bland and Altman limits of agreement were lower than 0.2 m/s, and the ARMSE was between 0.03 and 0.17 m/s depending on the model and the speed regime. For instance, considering a walking man of 3 km/h (0.8 m/s), the error of the first two previous models was between 6.7 and 10%, and that of [31] was between 3.8 and 21.2%. For a running show jumping horse of average speed 350 m/min (5.8 m/s), the presented model error of 0.43 m/s was about 5.2%. Thus, the present model could be considered as more accurate than most of the existing ones for human walking.

As a result, to pursue refining the accuracy of our model, more campaigns of measurement with the reference system are needed. As a matter of fact, a panel of 58 horses is not sufficient to model the behavior of all horses due to individual’s diversity. It is also necessary to do more measurements on curves of various diameters, since this greatly influences the horse’s behavior, the corresponding collected signals, and therefore, the horse’s speed [32]. In addition, more extreme speeds should be collected as well, in order to improve the model behavior in those particular cases. Indeed, slow and extremely fast canter strides represent currently less than 20% of the database, which could explain the lower model performance for those situations.

Despite its accuracy, our model presented some limits: we should validate the use of an automatic chronometer to estimate average speed in a future study, comparing it to the camera motion capture system on straight paths for example. Moreover, the presented model should not be directly transferred to a discipline other than show jumping because show jumping canter is specific to the discipline. For example, in endurance horses, flat canter is preferred, while in show jumping, the bounce is important in order to help the horse gather more vertical than horizontal energy to ease the jumps. Therefore, in order to adapt our tool to other disciplines, the model has to be expanded with more data gathered in new situations. The SVM model is transposable to the other equestrian sports and to bipedal locomotion, as long as consistent data are provided to train the model.

## Figures and Tables

**Figure 1 sensors-20-00518-f001:**
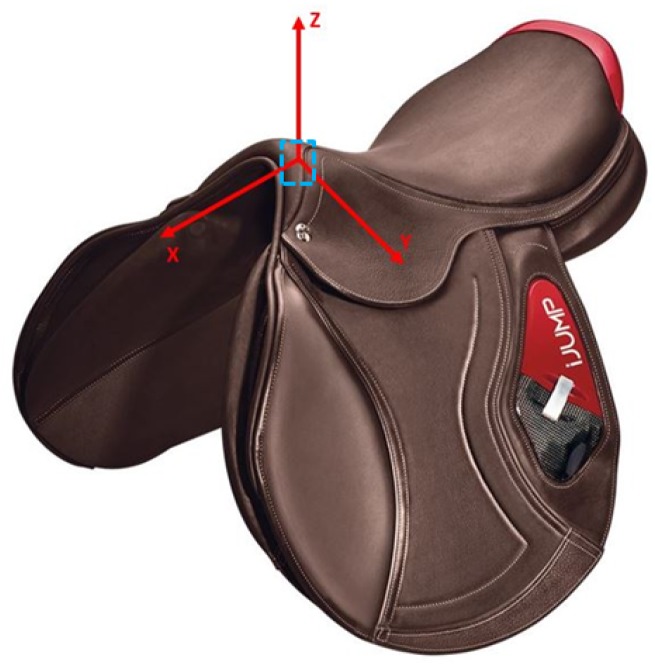
Orientation of the IMU’s axes and sensor location (blue dotted lines).

**Figure 2 sensors-20-00518-f002:**
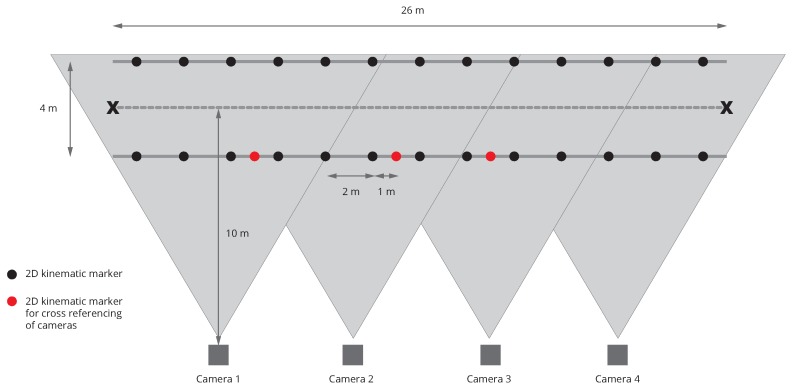
Field of measurement with 2D video cameras.

**Figure 3 sensors-20-00518-f003:**
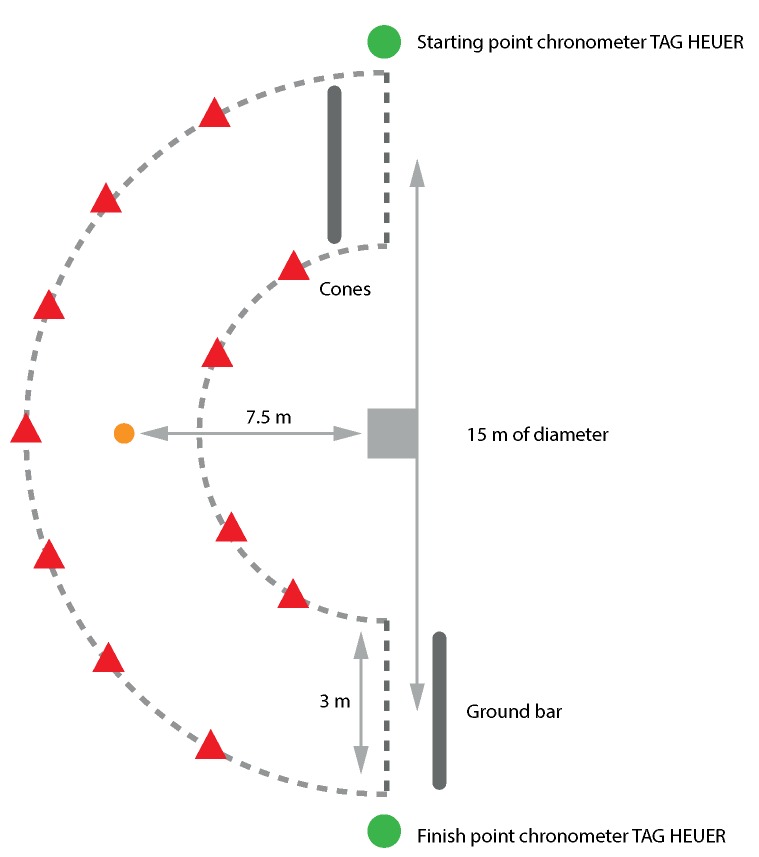
Plan of speed measurement on a curved path for a horse at left hand canter.

**Figure 4 sensors-20-00518-f004:**
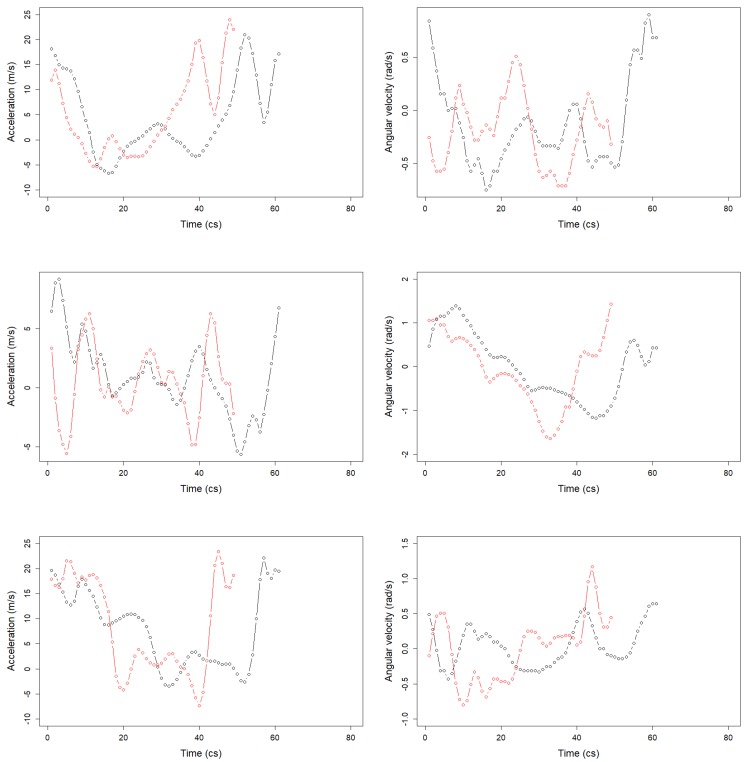
One stride acceleration (**left**) and angular velocity (**right**) signals for the *x*-axis (**top**), *y*-axis (**middle**), and *z*-axis (**bottom**). The red curve corresponds to a running speed of 8.6 m/s, and the black one corresponds to a running speed of 5.0 m/s.

**Figure 5 sensors-20-00518-f005:**
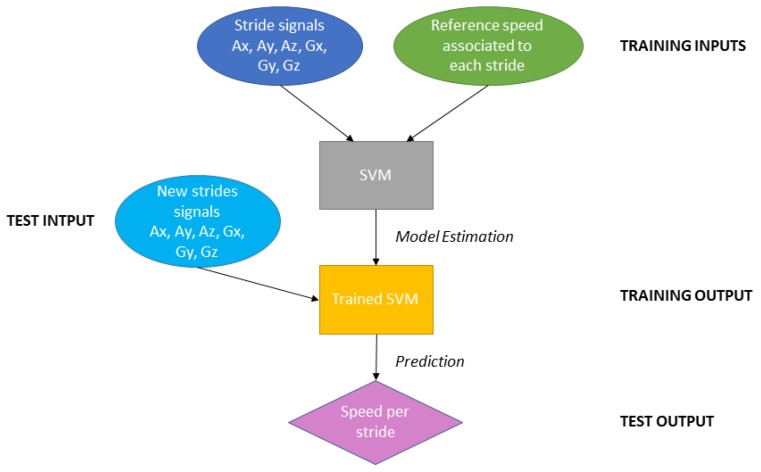
Diagram of the SVM process from training to the speed prediction.

**Figure 6 sensors-20-00518-f006:**
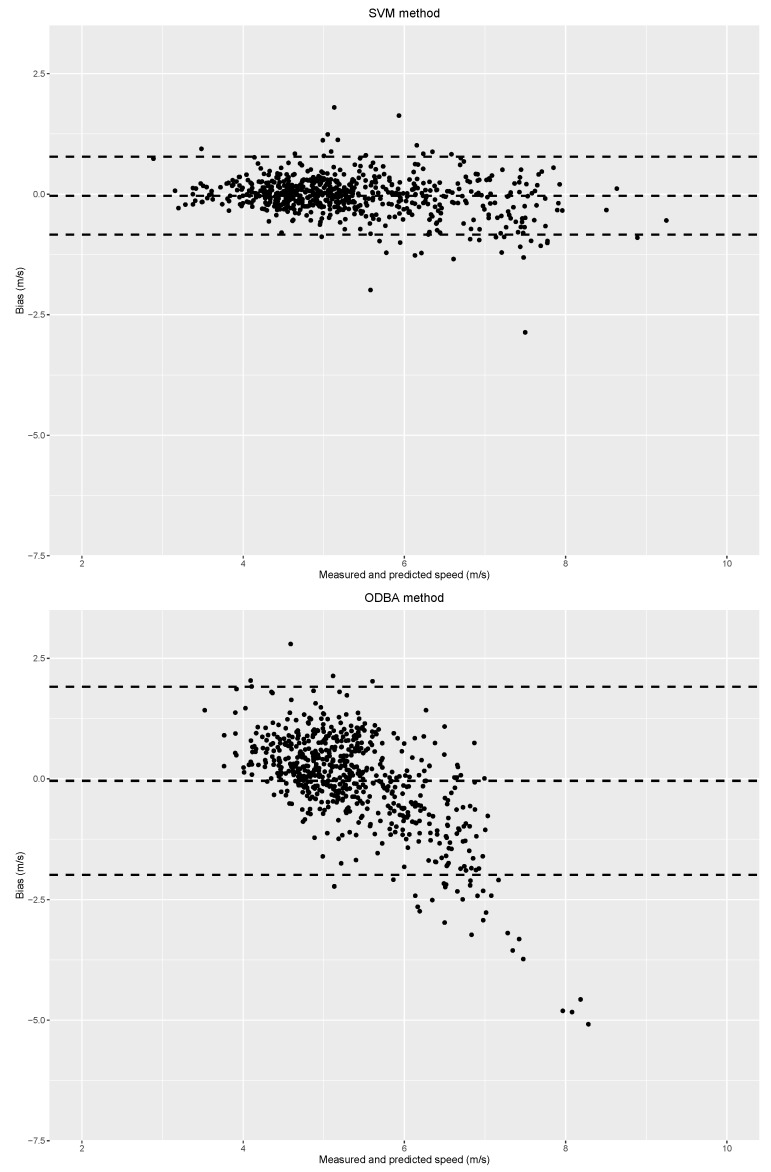
Bland and Altman plot for one repetition of the SVM model with its 95% confidence interval (**top**) and the overall dynamic body acceleration (ODBA) method (**bottom**).

**Table 1 sensors-20-00518-t001:** Mean, minimum, and maximum of the percentage of error above 0.6 m/s and the mean (standard deviation) of the width of the Bland and Altman limit of agreement for 50 repetitions for each method.

	SVM	ODBA
Mean of model error above 0.6 m/s	10.9%	51.4%
Minimum of model error above 0.6 m/s	9.0%	47.8%
Maximum of model error above 0.6 m/s	14.0%	55.1%
Mean of width of the limit of agreement	1.7 m/s	3.9 m/s
Standard deviation	0	0.1
Average RMSE	0.43	0.98
Standard deviation of RMSE	0.02	0.03

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
