# Peer review of "A Method to Estimate Horse Speed per Stride from One IMU with a Machine Learning Method"

_sensors, 2020, doi:10.3390/s20020518_

Round 1

Reviewer 1 Report

Overall comments to the authors:

The manuscript introduces the application of machine learning method to estimate horse speed per stride which has the potential to become a relevant research tool in equine biomechanics.

Please make sure that you scan through the manuscript for the statements which includes the word “Indeed” this has been quite repetitive and recommended to rephrase those sentences and paragraphs.

Formulation of “we” statements recommended to be replaced by the authors or in the present study etc. (Line 91, 137, 138, 156, 170, 187, 192, 208). The same recommendation could be extended to the formulation of “our model” to replace with e.g. the model of the present study.

Detailed comments:

In the introduction section, the authors well-presented the ongoing challenges with IMU sensors.

Line 62-64: The accuracy level has to be matched to measurement settings, therefore it would be ideal to hear from the authors what level of accuracy is necessary for a training session and during a competition event.

Line 80: Please describe exactly where and how the sensor was attached. Also, consider to include a photo or an image of the sensor location.

Line 88: Please explain your coordinate system e.g. Z-axis vertical. This information could be added to the figure legend as well.

Line 98: Manufacturer of the tracking system is not so easy to read, please follow the journal policy to list out the model number and other necessary manufacturer’s details.

Line 137: Please rephrase the sentence to “thus we do not care…”

In the discussion session, it would be great to read more about the under and overestimation of the technique (high speed vs. low speed), which was nicely explained in the results section of the manuscript.

Line 188: Please explain what Z-axis means in your coordinate system, this might mean different direction for readers (try to use, for example, vertical axis).

Line 229: Please explain what you mean with the term bounce.

Line 233: Please consider to rewrite this whole paragraph to increase readability, more precise details of this conclusion are needed.

Author Response

Dear Editor,

We took notice of both reviewers feedbacks, all changes made in the manuscript has been highlighted in red. You will find our answers in the pdf document.

Kind Regards.

Reviewer 2 Report

Authors developed an IMU-based method to estimate horse speed, yet there are major concerns about the clarity and validity of the methodology.  Please see the following specific comments.

Abstract:

While the Abstract is well-organized, there is still a lack of proper flow.  For example, the very last sentence seems confusing; what is data slicing; which good results?

Line 4:   It seems that the main goal of the study is to develop a method to measure the speed, not a smart device.

Line 7: What are these methods for? Why did you implement them? And how? Please briefly explain.

Line 8: Please be more specific here and try to introduce and report the quantitative measure (probably the accuracy or error rate) you have used to compare the methods. Line 164 and 165 or data from table 1 would be proper.

Line 11: why are you comparing the method developed for the horses with the methods developed for human gait analysis? Is that a valid comparison? Is it helpful to compare them? Here, we are talking about two inherently different gait types; this comparison is like to compare apples and oranges. 

Line 12: As mentioned earlier, which good results? It would be better to replace this sentence with something relevant to the application of the developed platform.

Keywords: “system” should be replaced by “support”!

Introduction:

Line 22-26: please break the sentence. This long sentence has made the content difficult to follow.

Line 35: you mentioned 3 methods. However, I just could see 2 methods introduced in this paragraph. Is the third method the one using the statistical methods?

Line 63: Regarding ‘his horse’, please replace “his” with “his/her”.

Method:

Line 88: Why 100 samples? How did you choose 100, not any other number?

Another concern is that why didn’t you consider the whole signal to be analyzed for stride identification?

What is the definition of stride for a horse from a physiological viewpoint?

Line 113: You have used two different and totally different method as the gold standard to measure the speed of the horses. How could you validate the speed estimation with a chronometer? What is the range of error for that? You could possibly do a validation study for this measuring system using your Camera motion capturing system.

Also, by calculating the speed per stride for a curved path, do not you think you are increasing the error of the model? How do you justify the assumption of having the average speed in all of the strides in a curved path?

Line 126: Why the cut-off frequency of 10 Hz? Please provide references to the literature.

Line 134: Did you mean “regression” instead of “prediction”?

Major concern: here at the core of the study. However, you wrapped it up in half a page. I want to see a sample of the data collected by IMU in a plot for a stride (all 6 measures). How does it look like?

In paragraph 4 of the Introduction section, you mentioned SVM is used for the classification. However, you did not point out here you are using the SVM for classification. I assumed that you have used it for classification and then implement another algorithm for regression. Please explain the approach, the labels, and the outcomes for both parts of the classification and regression. It is not clear what were the labels and outputs.

In case you just implemented SVM for regression, was the input IMU signal for each 100 sample points that you mentioned before? Why did you choose SVM for regression? Since your application is regression, why didn’t you try other approaches such as linear regression, lasso regression, and logistic regression?

Also, you have used all 6 measures for your algorithm. How would it be if you pick some of them? As an example, what if we had acceleration in the x-direction and angular velocity for a gyroscope in the y-direction. This could have conducted under the subtitle of feature extraction. Currently, it is not clear if all 6 measure have the same role in the calculation (prediction?) of the speed.

Line 146: What is the difference between the training set you mentioned earlier in line 138 and this one? You already mentioned 75% as the portion of the training set. Here you set it to 80%. Please further explain it.

Major concern: The definition of the error seems to be the mean error for all the data points with an error of more than 0.6. Notably, this measure is not dimensionless to be used as a percentage error. Furthermore, the idea to just consider the points with an error of more than 0.6 does not seem reasonable and valid. The error should have calculated for all of the data points regardless of their values. Finally, you could consider the 0.6 as your target error to be reached. You can check the literature on the model accuracy for the regression methods.

Line 177: Please redesign the table 1. It is confusing when you write a ‘minimum of error >0.6’. It would be better to write a minimum of the model error.

Discussion:

Line 180: Again you claimed to develop a smart device. However, you are implementing a previously developed system to measure the horse speed using your developed method.

Line 188: the stride is not detecting in the current study. Just the initial point is detected and picked along with the following 100 points.

Line 190-191: how did you claim that this approach will increase the accuracy? Have you done any study to compare the two methods of stride detection?

Line 204? Is this claim valid? How could you justify the developed model has an accuracy of 0.6?

Major concern: Line208-220: I went through the references that you have in this paragraph to compare your model with them. The nature of the errors for the current study and the references are totally different. The references 27 and 13 initially calculated the error (%) for each stride. Then, they have put the values in the statistical process to calculate the error which was ARMSE. This is a common method to measure the regression error. However, the error that we have in this manuscript is different in nature and does not seem to be valid.

Line 233: This is not valid for the labeling based on chronometer and assigning the average speed to all strides in a curved path.

Round 2

Reviewer 2 Report

The current version of the manuscript has addressed most of the earlier comments. However, there are still some concerns that need to be addressed.

The Abstract still need some modifications; 1) Please remove the sentence in line 9 &10; rather focus on measures of model performance such as accuracy. 2) Most of the Abstract is related to the introductory statements (lines 1-7) or conclusion aspects (lines 11-18). Rather, authors need to summarize them and focus on methodology/results obtained. Why using section “0”? Please update to “1” in line 21. Figure 4 needs to be improved. The height of the plots could be enlarged. If you just check the figure there are a big gap between each plot and above/below plots. You can also decrease the height of ticks for X-axes and take advantage of that again to increase the height of the plots. To be more clear, please make sure the height of the plots are at least equal to the length of Y-axis title (e.g. Angular speed (rad/s)) Line 146: please add the reference for non-parametric method Line 144-147: have you tried the other method and got the best results from SVM? If you have not, please just be accurate and state you have implemented SVM out of all of the mentioned methods and remove ‘we will only present here the one that gave147 the best results.’ Line 144-147: Please connect this paragraph to the next one. Line 167: Reviewer’s earlier comments about the nature and mathematics behind this formulation are not addressed yet. However, in this version, you have added the RSME which is more valid and more comprehensible. I do not agree to present this new version of the error in this study yet. In table 1: please use the full form of standard deviation in the table Line 203: I still do not believe that you have detected the strides. This is not accurate. The authors detected the max peak and picked the point along with the 100 following points. This approach depending on the horse speed may pick more or less than a stride. In fact, the endpoint is not defined according to the definition of a stride in this study. Line 226-241: If you do not use the formulation that I have mentioned my concern about it, you need to modify this and the following paragraphs accordingly.

Author Response

Please see attachment. Changes are highlighted in red in the manuscript

Round 3

Reviewer 2 Report

Upon reading through the authors' response letter and updated manuscript, my recommendation would be to accept the paper with a need for minor English editing.